

# Fishery catch records support machine learning-based prediction of illegal fishing off US West Coast

Jordan T. Watson[1], Robert Ames[2], Brett Holycross[2], Jenny Suter[2,3], Kayleigh Somers[4], Camille Kohler[5] and Brian Corrigan[6]

[1] Pacific Islands Ocean Observing System, University of Hawaii at Manoa, Honolulu, HI, United States of America
[2] Pacific States Marine Fisheries Commission, Portland, OR, United States of America
[3] Pacific Islands Fisheries Science Center, National Marine Fisheries Service, National Oceanic and Atmospheric Administration, Honolulu, HI, United States of America
[4] Northwest Fisheries Science Center, National Marine Fisheries Service, National Oceanic and Atmospheric Administration, Seattle, WA, United States of America
[5] neXus Data Solutions, LLC, Anchorage, AK, United States of America
[6] West Coast Division, Office of Law Enforcement, National Marine Fisheries Service, National Oceanic and Atmospheric Administration, Seattle, WA, United States of America

Corresponding author
Jordan T. Watson, jwat@hawaii.edu

## ABSTRACT

Illegal, unreported, and unregulated (IUU) fishing is a major problem worldwide, often made more challenging by a lack of at-sea and shoreside monitoring of commercial fishery catches. Off the US West Coast, as in many places, a primary concern for enforcement and management is whether vessels are illegally fishing in locations where they are not permitted to fish. We explored the use of supervised machine learning analysis in a partially observed fishery to identify potentially illicit behaviors when vessels did not have observers on board. We built classification models (random forest and gradient boosting ensemble tree estimators) using labeled data from nearly 10,000 fishing trips for which we had landing records (*i.e.*, catch data) and observer data. We identified a set of variables related to catch (*e.g.*, catch weights and species) and delivery port that could predict, with 97% accuracy, whether vessels fished in state *versus* federal waters. Notably, our model performances were robust to inter-annual variability in the fishery environments during recent anomalously warm years. We applied these models to nearly 60,000 unobserved landing records and identified more than 500 instances in which vessels may have illegally fished in federal waters. This project was developed at the request of fisheries enforcement investigators, and now an automated system analyzes all new unobserved landings records to identify those in need of additional investigation for potential violations. Similar approaches informed by the spatial preferences of species landed may support monitoring and enforcement efforts in any number of partially observed, or even totally unobserved, fisheries globally.

## INTRODUCTION

Illegal, unreported, and unregulated (IUU) fishing is a growing concern globally (*Sumaila et al., 2020*; *Long et al., 2020*; *Hosch & MacFadyen, 2022*). The full impacts of IUU fishing on economies and ecosystems are difficult to quantify, but recent estimates suggest that millions of metric tons of IUU catches are likely to occur each year, depriving fishers, communities, and countries of tens of billions of dollars in lost revenue and taxes (*Long et al., 2020*; *Sumaila et al., 2020*). Moreover, the uncertainty in the magnitude of species removals by IUU fishing (including legal but unreported activities; *Song et al., 2020*) is difficult to measure, and impacts on the management and the sustainability of target and non-target species are challenging to quantify (*Sumaila et al., 2020*). A common type of IUU fishing, which has led to large-scale adoption of vessel monitoring systems (VMS) globally, occurs when fishing is undertaken in prohibited areas or at prohibited times (*Aneiros, 2002*). Thus, the ability to resolve spatial and temporal fishing patterns is crucial to monitoring both legal and IUU fishing.

Commercial fisheries, both legal and illicit, often rely upon the assumption that a particular species can be persistently harvested from predictable locations. This assumption is fundamental to spatially-explicit management measures for target and non-target species (*e.g.*, *Hazen et al., 2018*; *Welch et al., 2020*). For example, marine protected areas are ideally implemented in locations where they are expected to yield the greatest and most persistent benefit for conservation or restoration of species, biodiversity, or habitats (*Pressey et al., 2007*). Similarly, fishers may intentionally avoid areas (or time-area combinations) that are considered to be hotspots for species with constraining quotas (*Abbott, Haynie & Reimer, 2015*). While recent work has argued for increasingly dynamic marine spatial planning for species (*Pons et al., 2022*), some species, like demersal groundfish species with deeper habitat preferences (*e.g.*, *Jacobson, Brodziak & Rogers, 2001*) are likely to exhibit more persistent home ranges and thus more static fishery distributions.

The predictability of certain species distributions suggests that by examining the composition of fishers' catches, researchers can uncover information regarding the spatially-explicit choices that were made during particular seasons or years. Several studies have utilized species-specific habitat fidelity to discriminate fishery targeting strategies or métiers based on catch composition analysis (*e.g.*, *Lewy & Vinther, 1994*; *He, Bigelow & Boggs, 1997*). *Langseth and Clover (2021)* found that species-specific catches were sufficiently associated with particular depths in a mixed-species Hawaiian bottomfish fishery such that, in the absence of depth data, species composition could help to identify disparate CPUE indices for assessment applications. By extension, the presence of certain deep water species in a catch could be used to indicate that fishers traveled at least a certain distance from a particular port (or distance offshore) to reach a preferred habitat or depth. Thus, the presence of certain species in landing records could be used to infer whether fishing occurred within management strata or jurisdictions with particular habitat or depth characteristics.

In the United States, fisheries governance and jurisdictions are typically divided into two primary spatial strata, federally-managed waters (3 to 200 nmi ($\sim$5.6 km–370.4 km)

from shore) and waters managed by individual states (within 3 nmi of shore). Off the US West Coast, some commercial vessels are permitted to target groundfish in either the farther offshore federal or the nearer shore state waters, while others possess permits to fish in both state and federal waters. Vessels that fish in federal waters may be required to have a federal fishing permit and are required to transmit their locations at fixed intervals *via* a VMS. Vessels that fish in state waters that do not possess a federal fishing permit and a VMS are thus prohibited from fishing in the farther offshore federal waters (see the Supplementary Section for more details on VMS requirements). Because state waters permits do not require vessels to carry a VMS, it is difficult to assess whether these vessels are only fishing within the state waters jurisdictions for which they are permitted. While vessels that are only permitted to fish in state waters typically do not report their vessel locations, they do report the amounts of each species they catch on any fishing trip. Thus, if the composition of landed catches could help to identify whether a vessel had fished in state *versus* federal waters, it could provide valuable support towards assessing compliance for spatial fishery regulations.

The motivation for this study was the difficulty of enforcing a VMS regulation that only applied to a portion of the fishers in this region (ie., those permitted to fish in federal waters); thus, we sought to determine whether an analytical solution could provide an alternative to expensive aerial or at-sea monitoring. We tested the hypothesis that catch data (*e.g.*, species compositions, catch totals, delivery port) reported on groundfish landing records could predict whether fishing occurred in state *versus* federal waters. Our framework relied upon the assumption that certain species would have enough fidelity to either the shallower nearshore (state waters) habitats or the deeper offshore (federal waters) habitats that machine learning models could differentiate the catch records from different locations. We first trained models using labeled data (observed fishing trips) to discriminate landings records that originated from federal-only *versus* state-only fishing trips. Second, we created an operational environment in which newly collected fishing trip records would be automatically analyzed to predict whether these trips occurred in state or federal waters. These predictions were delivered to fisheries enforcement investigators through a customized platform in which flagged fishing records were noted for subsequent manual review. While the actual operational system is confidential, we describe both the workflow for the operational system and the conceptual design. These tools could be generalized and used to identify and investigate potential IUU behaviors in other fisheries and especially those with partial at-sea monitoring.

## MATERIALS AND METHODS

### Data

Commercial fishing for groundfish using fixed gear occurs in both state and federal waters off Oregon and California, and in federal waters off Washington. Retained and sold fish species are recorded on a state landings record regardless of whether fishing occurred in federal or state waters. These landing records include information about the vessel that made the delivery, the weight and price paid for each species/species group, the delivery

port, the date, and in some cases the general area in which fishing occurred. All of this information is self-reported by the fishers and dealers. These landings records are required for all trips; a portion of trips are also documented by on-board fishery observers.

The West Coast Groundfish Observer Program (WCGOP) was established in 2001 and deploys trained field biologists (hereafter called observers) to monitor vessels at-sea in all commercial fishery sectors that retain or discard groundfish, including the fixed gear fleets that operate in state and federal waters off Oregon and California, and in federal waters off Washington. Fixed gear includes, hook and line (*e.g.*, longline, *etc.*), pots, traps, pole, and jig gear types. WCGOP selects a stratified, random sample of fishing trips on which to deploy observers. Observer deployment strata are based on port groups, with vessels assigned to a port group based on their previous year's landing locations (http://www.fisheries.noaa.gov/west-coast/fisheries-observers/overview-observed-west-coast-fishery-sectors#west-coast-fixed-gear%C2%A0). Annual levels of coverage range from 3 to 53 percent of landings depending on the fishing method and the year (*e.g.*, *Somers et al., 2021*). On-board observers collect and record data including catch amount and composition, discards, and fishing effort characteristics, such as location and amount of gear deployment and retrieval.

## Data preparation

Landing records, observer data, and other auxiliary fishery-dependent data were consolidated in the Pacific Fisheries Information Network (PacFIN) centralized data warehouse. The majority of the data integration, cleaning, and processing was performed within the PacFIN data warehouse and the analyses were conducted in Python (version 3.9.12), ArcGIS Pro (version 2.5), and ArcMap (version 10.2). We analyzed all landing records from observed trips between 2002–2019. The records were filtered to include only landings from fixed gear vessels for which the plurality of the landed weight was groundfish and to include only species for which the landed weight was greater than 10 kg. This isolated the landings to the fixed gear fleet and excluded rare species that would not provide value to the models. The aim was to determine whether the catch composition from landing records could be used to identify catch from state *versus* federal waters off Washington, Oregon, and California.

We used GIS software to identify observed trips where the vessel fished exclusively in federal waters or exclusively in state waters based on the gear deployment and retrieval locations. These spatial strata ("federal waters" *versus* "state waters") became the model labels for each of the landing records linked to observed fishing trips. We excluded observed trips during which vessels fished in both state and federal waters.

Landing records were pivoted on species/species groups to create a wide dataset which contained one row for each landing. Thus, each landing record consisted of the label, weights by species/species groups as columns, and auxiliary information, such as landing date, delivery port and vessel. Species/species groups not present on a given landing record were assigned a weight of zero. An additional variable, month-of-year, was extracted from landing dates to account for potential seasonality or intra-annual drivers of fisher behaviors. We then split the landing records into training (80%) and testing (20%) datasets for the

development and evaluation of the models and for variable subset selection. The splits were stratified by the labels to ensure both the training and testing datasets had the same label proportions. All numeric variables (*e.g.*, species weights) in the training data were scaled using min-max normalization and the same transformations were blindly applied to the testing data to avoid data leakage.

## Variable selection and model fitting

We selected random forests and gradient boosting, ensemble tree algorithms, for this study because they excel at classification tasks (*Brieman, 2001*) (implemented with the scikit-learn Python libraries (*Buitinck et al., 2013*)). Random forests, a bootstrap aggregation (or bagging) method, emphasizes variance reduction by combining the results of multiple classifiers trained on different sub-samples of the same dataset (*James et al., 2021*). Whereas, gradient boosting tends to reduce both model bias and variance. In boosting, modified versions of the original data are used to train models sequentially, with each model iteration attempting to compensate for the failing of its predecessor to minimize training errors (*James et al., 2021*). Both model frameworks approach a binary classification problem by generating a predicted probability. In our case, probabilities of at least 0.5 were classified as "Federal waters fishing" while probabilities less than 0.5 were classified as "State waters fishing". The proceeding methods are also available in the code supplement (https://github.com/rames72/ML-to-predict-Illegal-fishing-off-U.S.-west-coast.git).

The two models were first deployed to identify the most relevant variables (*i.e.,* species/species groups) from the landing records to predict whether fishing occurred in state or federal waters. The subset of variables was chosen through two strategies, first using sequential forward selection of the training dataset, and second, a machine learning technique called "variable of importances" (also known as impurity-based feature importances), which calculates a score for all the input variables used in the model. The higher the score, the greater the effect that the variable has on the model predictions. For this second strategy, we used only the variables that were chosen through the sequential forward selection process and only data from the training dataset. The stacking of these two strategies was to reduce the complexity of the models by only using the most consistently meaningful variables.

Sequential forward selection is a greedy search algorithm that starts with no variables, and then with each step forward adds one variable at a time until all variables are in the model (*James et al., 2021*). At each step, the variable that provides the most improvement to the classifier performance is added (*James et al., 2021*). The model accuracies were evaluated at each step forward using 10-fold cross-validation. After an exhaustive search, evaluating all the variables within the training dataset, the final selected variables for each model were those that produced the highest mean accuracy with the lowest standard error using the fewest number of variables. Selecting the fewest number of meaningful variables was important for consistent predictions with low variance. The full list of more than 100 candidate variables is included in supplementary code and data.

The variable of importances scoring was performed on the variables chosen by the sequential forward selection process. The importance of each variable was determined by

the mean decrease in impurity produced by the variable across all trees in the models. These scores provide a highly compressed, global insight into the models' behavior and reliance on each variable for making predictions (*Molnar, 2022*). Our goal was to reduce the selected variables to the most relevant variables for each model, the group of variables that together provided greater than 98% of the overall score for making predictions. To calculate the variables of importance scoring, the models were fit to all of the training data, but only using the variables chosen by the sequential forward selection process.

We again fit both random forests and gradient boosting models to the training dataset to tune the hyperparameters, using the variables from our subset selection process described above. This approach used 10-fold cross-validation and a random grid-search to tune hyperparameters based on the highest mean predictive accuracy score. Models with the best hyperparameters were fitted one final time to the selected subset of variables from the training data and then the models were evaluated on the testing data (holdout dataset) using a confusion matrix, as well as the receiver operating characteristic (ROC) curve and area under the curve (AUC). A ROC curve illustrates the trade-off between sensitivity and specificity of a binary classification model. The AUC provides a measure of performance across all possible classification thresholds and is a numeric measure of model performance (with an optimal value of 1).

Increasing concerns with fisheries models that rely on species spatial distributions are range shifts and short-term species responses to extreme environmental conditions, like the marine heatwaves that occurred off the US West Coast between 2014 and 2019 (*e.g.*, *Bond et al., 2015*; *Amaya et al., 2020*; *Weber et al., 2021*). To ensure persistence of model efficacy despite such inter-annual variability, our selected subset of variables was also used to fit the models whose training data iterated through all but one year, and were tested on the remaining year. For example, models were trained with data from 2002 - 2018 and tested on data from 2019, with this procedure repeated for each year as the holdout dataset (*i.e.,* leave-one-out model validation). As outlined above, training data were scaled using min-max normalization and the same transformations were blindly applied to the testing data (individual years). The trained models were fit to each year individually as a holdout.

**Operationalization.** A recent focus of improved analytical capabilities has emphasized not only methodological and technical improvements, but also the value of operationalizing data products for greater utility to downstream users (*Welch et al., 2019*). Such operationalization delivers automated and accessible data products to less technical users. Previous work by the PacFIN program created a confidential online management application that delivers reports to NOAA enforcement investigators through a user-friendly front end with a PacFIN data warehouse backend. For this project, a new report was created within the confidential online management application that merged available information for each of the landing records and appended our model-predicted probabilities that vessels fished in federal waters without the requisite permits and mandatory VMS. These model-predicted probabilities were obtained by fitting the trained models (*i.e., pickled models*) from the development environment to new unlabeled (*i.e., out-of-sample*) landing records in the production environment for years 2017 to present (Fig. 1). Similar to the model training, the data were first filtered to isolate the

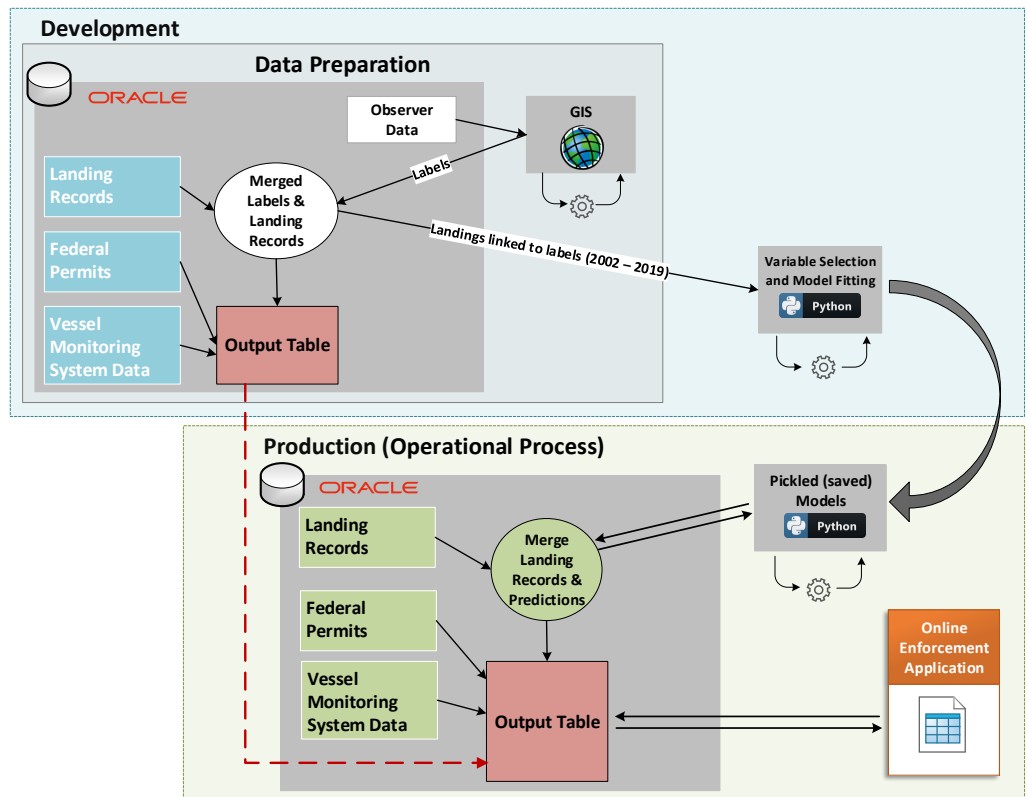

**Figure 1** **Conceptual diagram of the Model Development and Operationalized (automated) Project.** Labeled data are used in the development environment to train models, which are saved and used on new (unlabeled) data in the production environment. Model predictions and outputs from both phases are combined to support enforcement investigations *via* an online enforcement application.

landings to the fixed gear fleet for which the plurality of the landed weight was groundfish and to include only species for which the landed weight was greater than 10 kg. Predictive probabilities of at least 0.5 were classified as federal waters fishing while lower probabilities were classified as having occurred in state waters. These model-predicted probabilities of fishing locations could then help enforcement investigators to determine whether records may require additional manual review.

Enforcement investigators can infer several compliance-related behaviors from our management application, but the primary focus here is whether vessels without VMS or federal fishing permits were likely to have fished in federal waters. A record was flagged by our application if it met two criteria: (1) both of our models predicted that the retained catch on the landing record originated in federal waters (*i.e.,* probability at least 0.5); and (2) the random forest and gradient boosting models were in agreement and their predicted probabilities differed by less than 1.5%. We set this threshold low to force a high degree of precision among model predictions as an initial condition, but the thresholds for both the probability of fishing in federal waters ($p \geq 0.5$) and the agreement between the two models (<1.5%) can be easily adjusted based on input from enforcement analysts. Columns exist

in the final user report for the predicted probabilities from both models and the difference between the probabilities for each model. Thus, a user can sort and filter the landing records based on model predictions and precisions as they see fit.

## RESULTS

### Variable section and models

There were 9,225 observed trips (26,236 hauls) with 10,630 corresponding landing records from 2002–2019. These landing records included 113 unique species/species groups as recorded by the dealers from 703 unique fixed gear vessels in 61 ports along the US West Coast. We split these landing records into training and testing datasets, resulting in 8,504 (80%) training data with 53% labeled from federal waters and 47% from state waters, and 2,126 (20%) testing data with the same label distributions.

We deployed sequential forward selection that included all the 113 unique species/species groups found within the training dataset, as well as month-of-year and three engineered variables. The three engineered variables were, (1) total weight of catches on landing records, (2) total weight of all species landed per vessel per day, and (3) distance to 202 meter isobath from each delivery port. Through data exploration, we found that the total weight of catches on landing records were typically greater from federal waters trips than from state waters trips (Fig. S1). However, data from onboard observers revealed that vessels sometimes split their catches for a single trip across multiple fish buyers or dealers. During such split landing records, the individual landing records weights are thus less than the trip total weight would be, which could make the individual records appear less likely to have originated from federal waters, especially if none of the dominant species chosen as model variables were present. To accommodate such complexities, we included a second engineered variable, the total weight of all species landed per vessel per day (*i.e.,* across multiple landing records). The distance to 202 m isobath from each delivery port variable was created because our analysis revealed that sablefish (*Anoplopoma fimbria*), a deep water species, was primarily caught in federal waters (96.6% of records with sablefish catches; Table 1), but there were some exceptions. Along the California coast, there are a few ports where submarine canyons (*e.g.*, Monterey Canyon) occur in state waters. We estimated the distance from each port to the nearest sablefish fishing grounds and determined that 95% of observed effort targeting sablefish occurred deeper than 202 m (Fig. S2.). Port of trip origin was not consistently available, so this new variable, the "distance from each delivery port to the 202 m isobath", was added to models. This variable controlled for certain ports to more likely include legal catches of deep water sablefish within state waters. Month-of-year did not substantially improve model performance, likely because intra-annual effort distributions were relatively consistent across years (Fig. S3).

The sequential forward selection process produced optimal random forest and gradient boosting models using a subset of 10 (accuracy score $\mu = 0.976$, SE = 0.000861) and five (accuracy score $\mu = 0.973$, SE = 0.00129) variables, respectively (Fig. 2). Marginal increases in accuracy were achieved by adding more variables to the models as seen in Fig. 2, but at a cost of increasing the complexity of the models.

**Table 1  Numbers of observed landings records within state versus federal waters for the five species identified as most important in our classification models.** Tallies on each row count the number of records with at least 10 kg of a particular species and are independent of other species.

| Common name | Scientific name | Records from fishing in federal waters | Records from fishing in state waters | Total records w/presence | Proportion occurring in federal waters |
|---|---|---|---|---|---|
| Black rockfish | *Sebastes melanops* | 29 | 3,235 | 3,264 | 0.9% |
| Cabezon | *Scorpaenichthys marmoratus* | 3 | 1,006 | 1,009 | 0.3% |
| Lingcod | *Ophiodon elongatus* | 614 | 1,851 | 2,465 | 24.9% |
| Sablefish | *Anoplopoma fimbria* | 4,723 | 166 | 4,889 | 96.6% |
| Shortspine thornyhead | *Sebastolobus alascanus* | 1,860 | 133 | 1,993 | 93.3% |

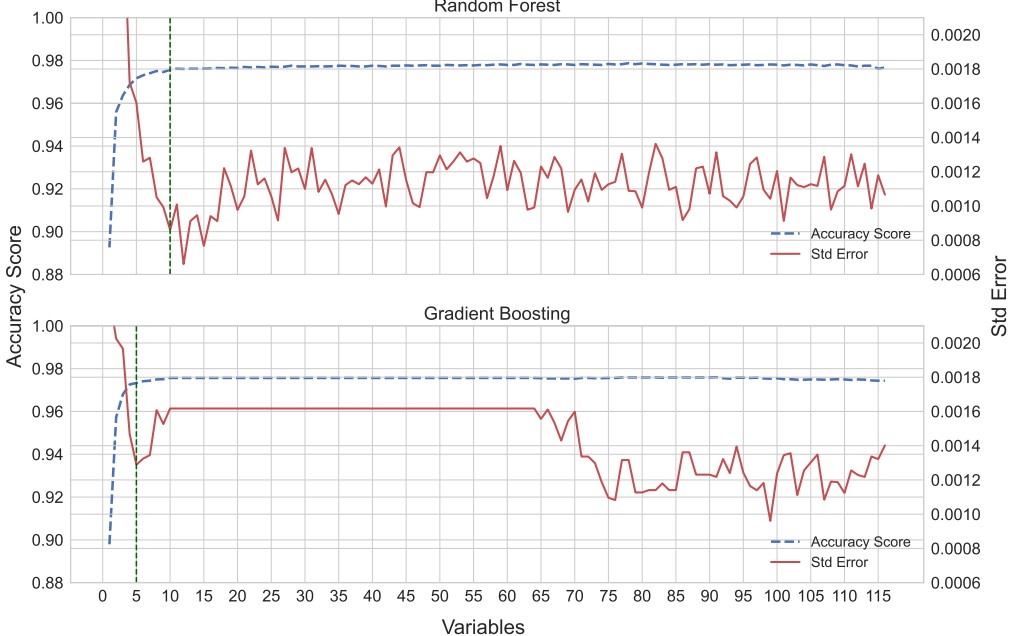

**Figure 2  Sequential forward selection process for the random forest and gradient boosting models.** The optimal model fit is shown by the vertical dotted lines, 10 and five variables, respectively.

We applied the variable of importances strategy to further refine the subset of variables from the sequential forward selection process to remove any irrelevant variables. This process yielded a smaller subset of seven variables for the random forest model. Those seven variables had a combined contribution of 99% of the model predictions. No variables were removed from the gradient boosting model (Fig. 3).

The five most influential variables were the same for both models, though the two models ranked the variable importances in different orders (Fig. 3). The five top variables included three species, sablefish, shortspine thornyhead (*Sebastolobus alascanus*), and black rockfish (*Sebastes melanops*), and two engineered variables, total weight of all species landed per vessel per day and distance to 202 m isobath from each delivery port. The three species had a high degree of fidelity to either state waters or federal waters with little variability,

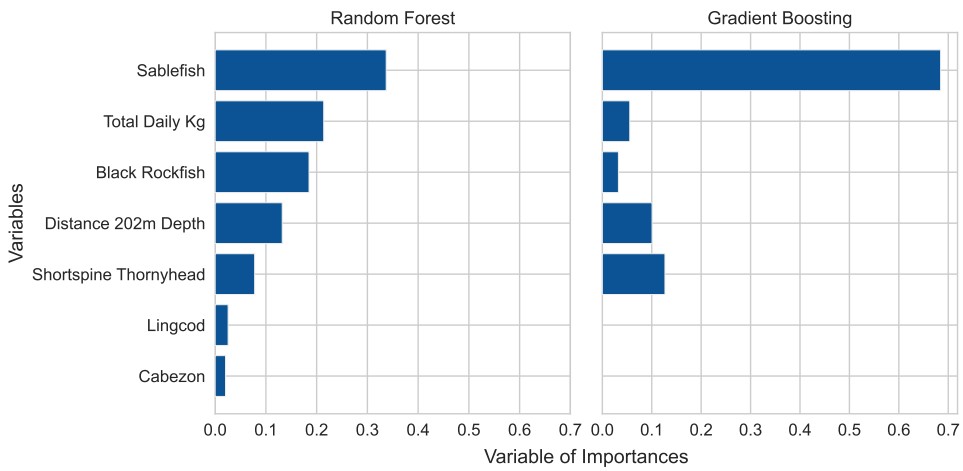

**Figure 3** **Variable of importances plots for the random forest and gradient boosting models based on the training data.** Note that the ranking of variable importances differed between the two models.

based on a close examination of the observed trips. The gradient boosting model relied mostly on sablefish, a species primarily caught in deeper federal waters, which accounted for nearly 70% of the overall importance score, highlighting the model's reliance on this species. Meanwhile, the random forest model relied more heavily on a mixture of deep and shallow water species along with the two engineered variables. Because the two models ranked these variables in different orders of dependence, suggesting different model fitting criteria, we retained both models as a method for redundancy and validation.

Initial modeling results yielded higher than expected errors (incorrect attribution of sablefish catches to federal waters) for eight ports. These eight ports were proximate enough to deep water that sablefish were more likely to be legally caught within state waters. While the distance to the 202 m isobath variable likely helped to reduce these errors, it was insufficient on its own. To adjust for this model bias and mitigate such errors, observed catches of sablefish in state and federal waters delivered to each of these ports were examined to determine the distribution of sablefish weights appearing on landing records. From such distributions, we identified a threshold weight for each port above which it was unlikely for catches to occur in state waters. As discussed above, the management application flagged a landing record as a potential violation based on two criteria, except for these eight ports. In these eight ports, when sablefish were on the landing record, the sablefish weight had to exceed the threshold value, in addition to the other two criteria, before the landing was flagged as a potential violation. For example, the port-threshold of sablefish on the landing record in Moss Landing, California had to exceed 481 kgs (Fig. S4).

The final trained models were evaluated on the testing data. Our diagnostics showed both models generalized well on the testing data, suggesting the models would also provide accurate predictions on the unlabeled or out-of-sample data (Table 2, Fig. S5). The normalized training and testing datasets used for this study are available online

**Table 2  Summary of random forest and gradient boosting classifiers based on the best hyperparameters that were fit to the training data, and generalized to the testing hold- out data.**

| Model performance | Random forest | Gradient boosting |
|---|---|---|
| Accuracy rate | 0.969 | 0.970 |
| Error rate | 0.031 | 0.030 |
| Precision | 0.971 | 0.971 |
| Recall | 0.971 | 0.973 |
| F1-score | 0.971 | 0.972 |

**Table 3  Summary of the random forest and gradient boosting classifiers based on the subset of chosen variables were fit using a leave-one-out approach in which all but one year was used to iteratively train models.** Rows denote the holdout (test) year for which model performance was evaluated.

| | Random forest | | | | Gradient boosting | | | |
|---|---|---|---|---|---|---|---|---|
| Year | Accuracy | Precision | Recall | F1-Score | Accuracy | Precision | Recall | F1-Score |
| 2002 | 0.97 | 1.00 | 0.97 | 0.99 | 0.97 | 1.00 | 0.97 | 0.99 |
| 2003 | 0.93 | 0.92 | 1.00 | 0.96 | 0.93 | 0.92 | 0.99 | 0.95 |
| 2004 | 0.95 | 0.95 | 0.95 | 0.95 | 0.97 | 0.97 | 0.96 | 0.97 |
| 2005 | 0.97 | 0.98 | 0.97 | 0.97 | 0.97 | 0.98 | 0.96 | 0.97 |
| 2006 | 0.95 | 0.92 | 0.95 | 0.94 | 0.96 | 0.93 | 0.97 | 0.95 |
| 2007 | 0.96 | 0.97 | 0.95 | 0.96 | 0.96 | 0.98 | 0.95 | 0.96 |
| 2008 | 0.96 | 0.96 | 0.97 | 0.97 | 0.96 | 0.96 | 0.98 | 0.97 |
| 2009 | 0.97 | 0.98 | 0.97 | 0.97 | 0.98 | 0.98 | 0.98 | 0.98 |
| 2010 | 0.97 | 0.97 | 0.98 | 0.98 | 0.97 | 0.97 | 0.99 | 0.98 |
| 2011 | 0.98 | 0.98 | 0.99 | 0.99 | 0.98 | 0.99 | 0.98 | 0.99 |
| 2012 | 0.98 | 0.99 | 0.96 | 0.98 | 0.98 | 1.00 | 0.96 | 0.98 |
| 2013 | 0.96 | 0.97 | 0.96 | 0.96 | 0.96 | 0.97 | 0.96 | 0.96 |
| 2014 | 0.97 | 0.97 | 0.96 | 0.96 | 0.97 | 0.98 | 0.96 | 0.97 |
| 2015 | 0.99 | 0.99 | 0.98 | 0.99 | 0.99 | 1.00 | 0.98 | 0.99 |
| 2016 | 0.98 | 0.99 | 0.97 | 0.98 | 0.99 | 1.00 | 0.98 | 0.99 |
| 2017 | 0.98 | 0.98 | 0.97 | 0.97 | 0.98 | 0.98 | 0.98 | 0.98 |
| 2018 | 0.97 | 0.97 | 0.98 | 0.97 | 0.98 | 0.97 | 0.98 | 0.98 |
| 2019 | 0.96 | 0.95 | 0.95 | 0.95 | 0.97 | 0.98 | 0.94 | 0.96 |
| Mean | 0.97 | 0.97 | 0.97 | 0.97 | 0.97 | 0.98 | 0.97 | 0.97 |

(https://github.com/rames72/ML-to-predict-Illegal-fishing-off-U.S.-west-coast.git). Raw landings records data are confidential and cannot be shared.

Given recent inter-annual environmental variability, such as marine heatwaves, it was important to ensure that model performance persisted across years. Iterative model training on all but one year demonstrated consistently high model prediction accuracy with each year (2002–2019) in the hold out (test) dataset (Table 3). The mean accuracy across all years for random forests and gradient boosting was 0.969 and 0.970, respectively. This relatively tight distribution of accuracies across years suggests that model performance was persistent over time, despite variable environmental conditions.

Ideally, training and test datasets will encompass the same unique feature values (*e.g.*, the same ports, same fish species) to avoid the potential for biased predictions. Our splitting of training and test datasets prioritized balancing the numbers of fishing trips for which we had observed data, and because not all ports had observed trips, not all ports appeared equally in the training and test datasets. This resulted in 10 unique ports that appeared only in the training data, three ports that appeared only in the test data, and 48 ports that appeared in both the training and the test data. Notably however, the port ID itself was not a model feature; individual port characteristics were incorporated into models based on their distance to the 202 m isobath. Those distances from port to the 202 m isobath that were exclusively within either the training or test datasets fell within the range of distances that were encompassed by values in both data sets. The robustness of our predictions indicate that any perceived imbalance did not impact model performance. Our annual leave-one-out model comparisons (Table 3) revealed consistent performance despite inter-annual variability in training and test data distributions. Additionally, 88% of records flagged for potential violations were for ports appearing in both the training and test datasets but there were also flagged records for ports that appeared in only the test data, suggesting that the model still encapsulated characteristics outside of the unique records within its training domain and mitigating some concerns for bias.

## Operationalization & reporting

Both trained models (random forest and gradient boosting) were applied to 57,401 unobserved (*i.e.,* out-of-sample) landing records from 1,557 unique vessels delivering to 92 ports and 748 dealers in years 2017 to 2023. Similar to model development, only fixed gear landing records where the plurality of the landed weight was groundfish were used, and minor species under 10 kgs were excluded. The models agreed on the location of fishing for 96.6% of the landing records, and the models and post processing yielded 564 potential violations from 143 vessels. Three vessels contributed to 172 (31%) of the 564 potential violations. After accounting for those three vessels, the distribution of flagged records across vessel sizes was consistent with the distribution of vessel sizes in the fleet. The mean potential violations (±SD) per vessel was 3.9 (9.38) with a median of 2, and flagged records were approximately uniformly distributed across months. Approximately 75% of flagged records occurred for ports with deep water (202 m isobath) within about 20 km, though importantly this distinction does not necessarily mean 20 km offshore (*e.g.*, deep water could occur in state waters but 20 km along shore from port). After controlling for the three vessels described above, flagged records were spread broadly across ports.

The records flagged as potentially fishing in federal waters represented substantially greater revenues than typical state waters fishing trips. Confidentiality rules prohibit in-depth details about the flagged records and vessels but we provide high level summaries here. The flagged records ($N = 564$) accounted for slightly more than $2M in revenue (an average of about $4,000 per trip) while 39,921 state waters records that were not flagged as potential violations landed about $26M (an average of $658 per trip). Among the flagged records, sablefish accounted for about 82% of the landed value whereas for the non-flagged records, sablefish accounted for only 5.3% of landed value. In fact, the total value of

sablefish in the flagged records exceeded $1.9M whereas the landed value of sablefish in the nearly 40,000 non-flagged state waters records was only $1.4M. Overall, these flagged records account for only 1.4% of the number of non-flagged trips but are equivalent to nearly 9% of the revenue for those trips, demonstrating the disproportionate impact that the flagged trips could have on the perceived value of the state waters fishery.

While our specific enforcement application is confidential, and thus cannot be shared, we illustrate the process by which our pickled (saved) models are stored and applied to new data as they are obtained by PacFIN and entered into the database (Fig. 1). As per the needs from end users (enforcement investigators), our model routine is automatically applied to newly submitted landings records once per month, and these data are merged with vessel attributes and fisher permits. All landings records with a potential violation are flagged and subsequently appear in a custom web application dashboard for manual inspection and further investigation by enforcement investigators (Fig. S6).

## DISCUSSION

Recent advances in fishery monitoring through technologies like VMS and Automatic Identification Systems (AIS) have improved the ability to track spatial fishing behaviors (*e.g.*, *Kroodsma et al., 2018*), but integrated data approaches are often still necessary to enforce fishing regulations (*Park et al., 2020*; *Suter et al., 2022*) especially as efforts to evade surveillance are also on the rise (*Welch et al., 2022*). Even fisheries with complete vessel surveillance must be monitored to determine compliance with vessel tracking regulations and some fleets are only partially monitored by vessel tracking systems, just as observer coverage is often limited to only portions of a fleet's effort. Such data gaps are often greatest for smaller vessels or those targeting nearshore species and the approach presented here could identify bad actors more readily or it may provide smaller vessels with an opportunity to demonstrate their compliance without additional regulations or onerous monitoring (*Song et al., 2020*). In this study, we used a supervised machine learning approach to analyze data from a group of vessels with partial VMS and partial observer coverage. We predicted, with consistently greater than 95% accuracy, whether vessels fished in offshore federal waters or nearshore state waters. Both of our classifier algorithms (random forests and gradient boosted ensemble tree classifiers) identified sablefish as the most important variable, followed by total catch weights and the weights of certain deep or shallow water species. The two models ranked variables in different orders of importance, leading to a multi-model approach that produces two predictions for each landing record. We successfully migrated our multi-model spatial-jurisdiction approach into an operational framework that analyzes new fishing records and appends model outputs to them, automatically flagging potential violations and expediting the review of large volumes of new fishery landings by enforcement analysts through a confidential online reporting tool.

### Model interpretation

Machine learning models often lack interpretability (*Rudin, 2019*). We have thus tried to explain our model inference *via* two relatively simple illustrations (Figs. 2 and 3). The

former of these two figures illustrates the interplay between predictive accuracy and the number of variables required to minimize prediction errors (note the relative stabilizations of accuracy and errors in Fig. 2). This approach enabled us to highlight that the prediction errors stabilized once the top seven variables were included in a model. Thus, while sablefish may have been the most important variable in both models (Fig. 3), additional variables were still necessary to achieve our optimization of prediction accuracies and standard errors.

Our application of these models to predict state *versus* federal waters fishing revealed intuitive variables of importance that are consistent with operational fishing characteristics and species distributions, though also driven by some inevitably complex interactions. The five species whose presence drove model performance were caught almost exclusively in either federal or state waters (Table 1). For example, out of 4,889 landing records with sablefish from observed trips that were designated as either federal or state waters, 96.6% fished only in federal waters. Meanwhile, thousands of landing records from observed trips in federal waters revealed that only 1% (at most) had black rockfish or cabezon. While simple data explorations may have revealed such persistent habitat or depth preferences for some species, more complex interactions between certain species distributions and trip-level catches or the bathymetry around certain ports may have been more difficult to identify manually or with traditional statistical methods. This is evidenced by the improvements that the 202 m isobath feature had on model accuracy, especially around certain ports. These interactions are further evidenced by the two different classification algorithms ranking variables differently but achieving similar levels of accuracy. Variable importance plots are not generally as interpretable as coefficient values in traditional statistical models, but the selection of the same variables across our classification frameworks emphasizes the ecological significance of our selected variables. Finally, while certain species may be caught almost exclusively in federal or state waters, it does not necessarily mean that species is present on all landings records from that region. Thus, it is not surprising that our models selected multiple species to inform their predictions, even if certain species, like sablefish, drove the models more than others.

Species catches and weights were not the only important variables selected by the models. For example, state waters are periodically quite deep (*e.g.*, in Monterey Bay) but total catch weights in state waters tend to be less than in federal waters. Thus, additional variables like "distance to the 202 m isobath" and "total catch weight" helped to control for some port-specific model interactions that might otherwise bias model predictions (*e.g.*, if sablefish were legally caught in state waters). Through the operationalized routines we present here, more complicated species distributions in some ports may lead enforcement investigators to choose different model prediction thresholds or to place greater scrutiny on potential violations that are flagged for certain ports, like those with access to deeper waters nearshore.

Models that seek to characterize fishery spatial distributions in the current era must also consider how environmental variability may affect species distributions and subsequently, the fishing activities that target those species. If changes in the environment lead fish and fishers to occupy different space (*Rogers et al., 2019*), jurisdictional boundaries may

be increasingly crossed by fleets, fishers may be motivated to fish in new or different areas (legally or illegally) (*Pinsky et al., 2018*), and the efficacy of model predictions may change or diminish. Our study area included a region that has experienced unprecedented extreme marine heatwave events in recent years (*e.g.*, *Jacox et al., 2018*) that have driven species redistributions (*Welch et al., 2023*). For some deep water demersal species like sablefish, such extreme events may be less likely to affect the behaviors of adult fish in ways that alter fishing locations in the short term, but these dynamic conditions may impact recruitment for some species (*e.g.*, *Tolimieri et al., 2018*) and subsequently, lead to longer term impacts on species distributions (*Shotwell et al., 2022*). Meanwhile, for nearshore species (*e.g.*, those found primarily in state waters), temperature-mediated shifts to deeper waters (typically farther offshore) have been documented (*Dulvy et al., 2008*), though the interactions among temperature, oxygen, stratification, and other oceanographic factors may complicate species- and region-specific responses (*Keller et al., 2015*). Given such concerns, we tested our models to determine whether years with anomalous environmental conditions impacted our prediction of state *versus* federal waters fishing with each of 18 years used as individual holdout (test) datasets. Cross-validation results were remarkably consistent (Table 3), with average accuracies around 97%. Despite our robust model performance, our framework for both training and operationalizing these models is straightforward so future years can be easily tested with newly observed fishing trips and model performance can be continually evaluated to assess shifts in fisher behavior or model accuracy.

## Application for enforcement

Initial use of our operational system by enforcement investigators revealed important distinctions around the types of potential violations in this fishery. It is easy to assume that fishery violations result from intentionally illicit behavior but the majority of the investigations prompted by our system thus far have revealed simple and easily correctable situations. For example, trips were out of compliance when a vessel had a VMS that was not turned on, that was malfunctioning, or that was not transmitting at the prescribed interval, resulting in federal waters trips without VMS data. These cases often occurred by vessels whose trips were usually in compliance, suggesting that the violation was unintentional. Additional cases emerged where a vessel was transmitting VMS data but their fishing permit had expired and our system correctly flagged their landings records for fishing in federal waters without a permit. Thus, while an important additional field of study is the motivation that leads fishers to intentionally break the rules, such illicit behavior may prove to be in the minority of violations in this fishery. Our system is setup to evaluate compliance based on clearly established, objective criteria and to infer management actions that might alter fisher motivations is beyond the scope of our study. Nonetheless, because our study revealed that many violations were readily corrected through relatively mechanical aspects of VMS systems or their data transmissions, it may benefit NOAA to assess whether an investment in the VMS program may bolster compliance and free enforcement investigators to explore the less common but inevitably more complicated cases of intentional violations.

There has been increasing attention on how seemingly objective criteria applied to model outputs or biases implicit within machine learning models themselves may lead to harmful, inequitable, or unintended outcomes (*O'Neil, 2017*). Drawing from the literature on terrestrial law enforcement or criminal justice more broadly, where algorithmic approaches have a longer history, numerous studies have sought to explore concerns around transparency (*e.g., Mittelstadt et al., 2016*), bias, and harm reduction (*e.g., Altman, Wood & Vayena, 2018*). *Llinares (2020)* presents a philosophical exploration of predictive policing and ultimately recommends a "critical and informed" view of such algorithmic approaches over a technophobic view. Importantly, *Llinares (2020)* stresses the use of empirical data to inform one's perspective while also understanding any biases that might be associated with those empirical data. This resonates with the harm-reduction framework proposed by *Altman, Wood & Vayena (2018)* that emphasizes trying to understand how an algorithm may disproportionately affect one group more than another. In fisheries, labeled training datasets are often small because the field is characterized by rare events (*e.g.*, catches of certain bycatch species) or difficult behaviors to capture, highlighting a potential source of bias. For example, *McDonald et al. (2021)* demonstrated the noble use of AIS data to identify vessels suspected of human rights abuses at sea, but *Swartz et al. (2021)* expressed concerns over the small number of relevant AIS profiles for the proposed vessel group and an imbalance in labeled data. In our own study, we scrutinized potential sources of bias and intentionally omitted any vessel characteristics (*e.g.*, size) or previous interactions with law enforcement that might unfairly flag certain vessels, permit holders, ports or other groups. Our model indirectly includes the fishing port (port distance to the 202 m isobath) and despite having about 10,000 labeled trips in our model, not every port was included in our training data because not every port had observed trips. This imbalance in our data could potentially lead to a bias in our models, though the distances to the 202 m isobath for the missing ports fall within the range of ports distances for which we have labeled training data. Thus, while we considered potential biases in our approach, it may be impossible to completely avoid them and due diligence is important in considering where inequities might occur and how they might affect particular groups (*e.g.*, certain vessel sizes or vessels fishing from a certain port).

The models we developed and operationalized for enforcement provide a tool that supports a greatly expedited first review of large amounts of data. For example, out of nearly 60,000 landings records, our models identified 564 records (*i.e.,* potential violations) for which further review was suggested. The manual review of even 564 records could be labor intensive, but 60,000 records would be simply unrealistic. Furthermore, since our analyses revealed that three vessels accounted for a third of potential violations, enforcement investigators may use this information to more intently scrutinize these vessels or to identify additional controlling factors that could be introduced into our models to exempt certain behaviors (*i.e.,* removing bias if those vessels seem inappropriately flagged). This approach is similar conceptually to the use of deep learning algorithms in review of electronic monitoring footage in some fisheries, where algorithms first reduce the amount of video requiring further review by identifying frames during which fishing events are occurring (*Qiao et al., 2021*). In our case, we have built models that are connected to

an application behind a confidential firewall that allows authorized data users to more efficiently identify suspect records from the entire dataset rather than a random subset of the records. Importantly, our approach does not lead to an algorithm issuing citations to fishers but simply identifies records that warrant a more detailed review by human investigators.

## CONCLUSIONS AND FUTURE WORK

Our analyses relied on a supervised machine learning approach with more than 10,000 labeled landing records (*i.e.,* observed as state *versus* federal waters fishing), but an unsupervised machine learning approach may also be applicable for similar efforts to stratify fishing activities based on catch compositions. In particular, because we observed such strong fidelity of certain species to either nearshore (*e.g.*, cabezon, black rockfish) or offshore (*e.g.*, sablefish, shortspine thornyhead) habitats, future work or work in other regions may find models capable of clustering many landing records into groups based on landings information without having to use labeled data. For example, *Brownscombe et al. (2020)* obtained similar results when comparing supervised random forests with unsupervised fuzzy k-means clustering algorithms to characterize spawning sites for a marine fish species. Thus, while many studies have focused on identifying IUU fishing primarily through vessel tracking systems like VMS or AIS, machine learning-based analysis of landings records may offer substantial promise in some cases with scarce or no at-sea data. However, as the previous section highlighted, caution should be used with unsupervised approaches to ensure that a lack of validation data does not unfairly bias inference against certain groups or individuals.

Several common data challenges in fisheries management are linked to limited coverage by observers or vessel tracking systems, as well as the disconnect between research-level analysis of fishery-dependent data and near real-time needs for analytical products by end-users (*e.g.*, managers and enforcement investigators). This study developed a valuable and novel analytical product while also building the data infrastructure to serve the outputs to its end users. Further, this work relied on an innovative and explicit collaboration among researchers, data scientists, and enforcement officers and analysts. This end-to-end approach merged multiple unwieldy datasets to connect a discreet analytical and operational need. With the proliferation of machine learning approaches, improved data management systems, and more collaborative frameworks, some of the traditional limitations of partial monitoring coverage and big data may become increasingly surmountable for more sustainable fisheries management.

## ACKNOWLEDGEMENTS

Previous versions of this manuscript were greatly improved by comments from M Hunsicker, J Langan, A Mamula, B van Poorten, H Welch, and one anonymous reviewer. Additional thanks to R Ryznar for administrative project support.

### Funding

Camille Kohler (neXus Data Solutions LLC) was paid under contract with Pacific States Marine Fisheries Commission. The funders had no role in study design, data collection and analysis, decision to publish, or preparation of the manuscript.

### Grant Disclosures

The following grant information was disclosed by the authors:
Pacific States Marine Fisheries Commission.

### Competing Interests

Camille Kohler is employed by neXus Data Solutions LLC.

### Author Contributions

- Jordan T. Watson conceived and designed the experiments, analyzed the data, prepared figures and/or tables, authored or reviewed drafts of the article, and approved the final draft.
- Robert Ames conceived and designed the experiments, performed the experiments, analyzed the data, prepared figures and/or tables, authored or reviewed drafts of the article, and approved the final draft.
- Brett Holycross conceived and designed the experiments, performed the experiments, prepared figures and/or tables, authored or reviewed drafts of the article, and approved the final draft.
- Jenny Suter conceived and designed the experiments, authored or reviewed drafts of the article, and approved the final draft.
- Kayleigh Somers conceived and designed the experiments, analyzed the data, prepared figures and/or tables, authored or reviewed drafts of the article, and approved the final draft.
- Camille Kohler conceived and designed the experiments, prepared figures and/or tables, authored or reviewed drafts of the article, designed the web application, and approved the final draft.
- Brian Corrigan conceived and designed the experiments, authored or reviewed drafts of the article, and approved the final draft.

### Data Availability

The fishery landings data are included as covariates, broken into training and test data sets, so the exact data splits can be used for modeling. The model label data, as separate training and test data sets, and the Python Jupyter notebook with code are available in the Supplemental Files.

The code and data are also available at GitHub: https://github.com/rames72/ML-to-predict-Illegal-fishing-off-U.S.-west-coast.

## Supplemental Information

Supplemental information for this article can be found online at http://dx.doi.org/10.7717/peerj.16215#supplemental-information.

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
