# Peer review of "Fishery catch records support machine learning-based prediction of illegal fishing off US West Coast"

_PeerJ, doi:10.7717/peerj.16215_

## Round 0.1 · original submission · Minor Revisions

As you can see, two reviewers commented on your manuscript. Their comments are generally very positive, suggesting only specific or minor issues to be addressed.

Reviewer 1 ·

Basic reporting

The article is well-written, clearly reported, and technically sound. The hypothesis is well-defined and thoroughly investigated. My chief concern, is that this is an overtly technical approach to what is a fundamentally human problem. What incentives might exist for the "potential violations" that the authors describe, how have these incentives changed over the years, and how might these additionally be addressed in addition to issuing fines and citations? While I can accept the enforcement "stick" rather than the management "carrot" to be the main focus of this analysis, the other approach should at least be explored in the discussion. Relatedly, what is the logical end-point of management approaches and enforcement action primarily based on big data, machine learning, and artificial intelligence? While these may be valuable tools, they are not without risk in perpetuating bias an inequity (as has been documented in other forms of policing and as the authors briefly allude to towards the end of the manuscript). Indeed fisheries management is fundamentally a human process based on relationships and efforts to reconcile divergent perspective to reach consensus. I've copied some suggestions for citations that could be incorporated into more elaborate discussion paragraph(s), that I think might provide valuable context. As type of approach the author's are suggesting is rapidly gaining popularity in the scientific literature and management fields, I think it is important to sound a note of caution and acknowledge the trade-offs.

Suggested References:

-Swartz, W., Cisneros-Montemayor, A.M., Singh, G.G., Boutet, P. and Ota, Y., 2021. AIS-based profiling of fishing vessels falls short as a “proof of concept” for identifying forced labor at sea. Proceedings of the National Academy of Sciences, 118(19), p.e2100341118.
-Scoville, C., Chapman, M., Amironesei, R. and Boettiger, C., 2021. Algorithmic conservation in a changing climate. Current Opinion in Environmental Sustainability, 51, pp.30-35.4
-Mittelstadt, B.D., Allo, P., Taddeo, M., Wachter, S. and Floridi, L., 2016. The ethics of algorithms: Mapping the debate. Big Data & Society, 3(2), p.2053951716679679.

Experimental design

The author's technical expertise is evident in their thoughtful and well-documented approach.

Validity of the findings

Overall the findings appear sound. I have some minor requests for additional information and clarification (see below) but understand the challenging position of the researchers in not wanting to "tip their hand" to the extent that the effectiveness of the tool would be diminished.

Additional comments

Line 86-88: Can you provide some more information on US West Coast Fisheries Vessel Monitoring requirements? How recently were these regulations passed, under what authority, and how have they been revised over time? How does AIS tracking fit into the mix? Is there a size (i.e., length) limit beyond which a different set of rules apply? What are the incentives not to use vessel tracking technology (i.e., cost, privacy concerns)
Line 91: Aren’t at least some fishing vessels required to submit spatially explicit logbook information and/or coarse resolution “reporting blocks”?
Line 114: Why are there no state landings off Washington?
Line 114: What was the motivation to focus on this specific fishery. Did the researchers have an a priori hypothesis (if so what informed it?) or a reason to prioritize a fixed-gear, groundfish investigation or was this just selected as a test case due to data availability?
Line 125: Can you clarify how the sample is stratified? By gear, by target species, by vessel size, by Port all of the above?
Line 136: At first mention, clarify what is meant by fixed gear (i.e., traps, bottom-set longlines, both?)
Line 150: “Species/species groups not present on a given landing record were assigned a weight
151 of zero”. For fishing trips were there are both observer and landings records, what is the level of congruence between the two data sources (i.e., are there discards at sea that are noted by the observer but not registered on the landings receipt or is all catch landed?)
Line 320: What characteristics common across the 8 ports would the researchers hypothesize are responsible for these distinct dynamics (i.e., are they all in the same region, are they all of the same size, etc.)
Line 344: I realize that authors don’t want to tip their hand, diminishing the effectiveness of the tool, but it would help me evaluate the efficacy of the approach if there were some more descriptive statistics concerning what trips labeled as potential violations (i.e., Were most vessels ported in the same harbor? Were most trips undertaken by vessels of a certain size? Were most trips undertaken during a certain time of year? )
Line 346: What percentage of total catch value was represented by the 564 violations? This seems like important information to know how widespread the problem is, what impact it could have on the resource, and how much resources should be allocated to addressing it moving forward?
Line 362: Different regulatory standards, operating strategies, and incentives may exist for smaller vessels. Important to recognize this unique context so as to avoid criminalizing all small-scale operators.
See:
Song, A.M., Scholtens, J., Barclay, K., Bush, S.R., Fabinyi, M., Adhuri, D.S. and Haughton, M., 2020. Collateral damage? Small‐scale fisheries in the global fight against IUU fishing. Fish and Fisheries, 21(4), pp.831-843.
Line 450: More discussion is needed concerning how AI and ML approaches may function to enhance existing biases and inequities. Also, what are the possible negative externalities in terms of fostering trust and collaboration between fishers and resource managers inherent to this sort of surveillance approach?
Line 473: Were fishers engaged in addition to enforcement agents during any stage of the project? What might fishers suggest as the principal difference between landings in state and federal waters? Again, without tipping the hand of the enforcement folks, is there any information the authors can provide to ground-truth their findings?

·

Basic reporting

The reporting is very, very good. The authors clearly describe their methods in readable language. Figures and tables are well designed, though the supplementary figures do not have any figure titles that I could see.

Experimental design

No comment

Validity of the findings

The authors have gone to extreme lengths to ensure the parameter estimates from their models are robust - they included two different algorithms, two methods of variable selection and employed 10-fold cross validation to ensure their results were consistent.

The only issue I saw was that the test data had many more ports than the training data, which suggests there is the potential for there to be consistent differences between the two datasets that may bias prediction results. Were there consistent differences among ports in the two datasets that may lead to biased model predictions? Were there more/fewer predicted violations in ports that were included in both datasets compared to those from only the prediction dataset? This was never investigated, but certainly has implications for their operationalization.

Additional comments

line 400: you could remove one 'that' with no impact on clarity
There are obviously interactions among variables that drive model selection, yet are rarely discussed in the paper (beyond rationalizing the creation of novel variables like distance to 202m). I don't think these models predict these interactions, but would be useful to discuss a bit.
Findings like that related to the three boats that disproportionately lead to predicted violations were fascinating and really drive home the importance.

---

## Round 0.2 · accepted · Accept

Congratulations! I think you did a great job addressing the comments from the reviewers